# Early detection of new pandemic waves. Control chart and a new surveillance index

**Marta Cildoz, Martin Gaston[ID], Laura Frias, Daniel Garcia-Vicuña, Cristina Azcarate, Fermin Mallor[ID] ***

Institute of Smart Cities, Public University of Navarre, Campus Arrosadia, Pamplona, Spain

* mallor@unavarra.es

## Abstract

The COVID-19 pandemic highlights the pressing need for constant surveillance, updating of the response plan in post-peak periods and readiness for the possibility of new waves of the pandemic. A short initial period of steady rise in the number of new cases is sometimes followed by one of exponential growth. Systematic public health surveillance of the pandemic should signal an alert in the event of change in epidemic activity within the community to inform public health policy makers of the need to control a potential outbreak. The goal of this study is to improve infectious disease surveillance by complementing standardized metrics with a new surveillance metric to overcome some of their difficulties in capturing the changing dynamics of the pandemic. At statistically-founded threshold values, the new measure will trigger alert signals giving early warning of the onset of a new pandemic wave. We define a new index, the weighted cumulative incidence index, based on the daily new-case count. We model the infection spread rate at two levels, inside and outside homes, which explains the overdispersion observed in the data. The seasonal component of real data, due to the public surveillance system, is incorporated into the statistical analysis. Probabilistic analysis enables the construction of a Control Chart for monitoring index variability and setting automatic alert thresholds for new pandemic waves. Both the new index and the control chart have been implemented with the aid of a computational tool developed in R, and used daily by the Navarre Government (Spain) for virus propagation surveillance during post-peak periods. Automated monitoring generates daily reports showing the areas whose control charts issue an alert. The new index reacts sooner to data trend changes preluding new pandemic waves, than the standard surveillance index based on the 14-day notification rate of reported COVID-19 cases per 100,000 population.

**Data Availability Statement:** All the data are available at https://cnecovid.isciii.es/covid19/#documentación-y-datos.

## Introduction

The pandemic preparedness guide, published by the World Health Organization (WHO) [1], uses a six-phased approach to provide a framework to assist countries with pandemic preparedness and response planning. Phase 5, which extends throughout the post-peak period, focuses on addressing the health and social impacts of the pandemic, while preparing for

**Funding:** All authors of the article are recipients of the grants PID2020-114031RB-I00 (AEI, FEDER EU) and SISCOVID (Ref: 0011-3638-2020-000006). The funders had no role in study design, data collection and analysis, decision to publish, or preparation of the manuscript.

**Competing interests:** The authors have declared that no competing interests exist.

possible future waves of infection. Once the level of disease activity drops, there is no certainty whether more waves will occur and therefore surveillance is needed to keep health authorities informed of such possibility. Regions and whole countries have experienced successive waves of COVID-19 infection and thereby learned that any delay in the introduction of non-pharmaceutical measures at the outbreak and/or an immediate return to normal once it is over can result in a public health risk.

Following WHO recommendations, once one wave of the pandemic has subsided, public health administrations monitor the spread of the virus in order to brace for another. Countries around the world use a tiered system to determine when, where and to what extent COVID-19 restrictions need to be imposed. This is usually a color-coded system which goes from green, for areas with low case counts, to dark red, for those with high regional transmission levels. Currently, the main indicator used by the European Center for Disease Prevention and Control is the 14-day notification rate of newly reported COVID-19 cases per 100,000 population, an index we denote by $I_{14}$, which provides an estimate of the prevalence of active cases in the population [2]. Fig 1 shows the color code used to classify the European regions. It is defined by two indicators: the positivity rate and the $I_{14}$ index [3]. When deciding whether to introduce restrictions on free movement, the European Union established a coordinated approach, applying common criteria and thresholds based on pre-agreed color-coded COVID-19 transmission risk levels [4]. High and very high risk thresholds, coded in red and dark red, respectively, have changed with the evolution of the pandemic and vaccination levels. Usual values range from 75 to 150 for the red coded lower threshold and around 300 for the dark red coded lower threshold [4].

However, although commonly used metrics, such as the $I_{14}$ index and color-coded mapping, help in understanding current percentages of COVID-19 infections in the population, they do not describe the dynamics of the pandemic. Seroprevalence studies have also been proposed as a means to measure cumulative incidence [5], however, the results are subject to potential bias due to several factors such as vaccination, the lack of a representative sample of the target population, nonrandom willingness to participate in a survey, and failure to account for false negative serologic tests [6]. Furthermore, these studies are not carried out regularly enough to yield daily reports describing the real-time status of the population.

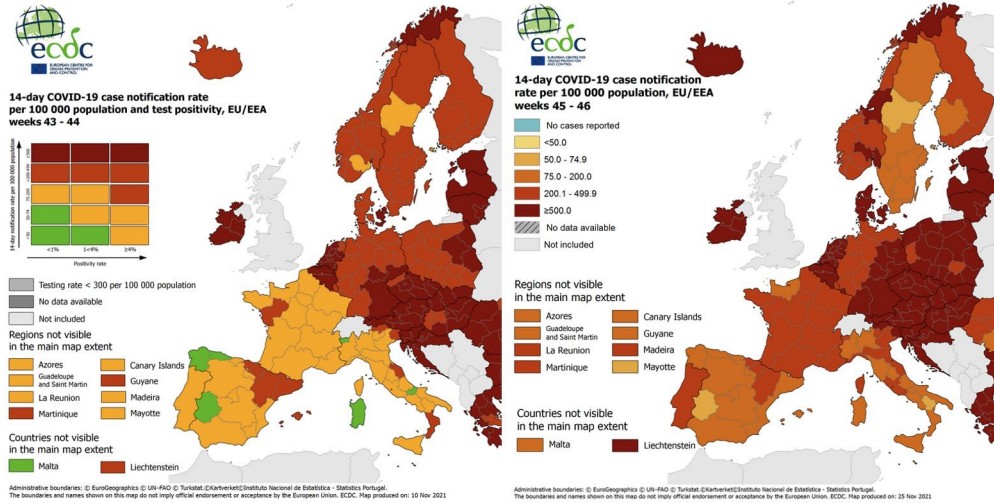

**Fig 1. Color-coded maps elaborated by European Center for Disease Prevention and Control.** https://www.ecdc.europa.eu/. Reprinted from https://www.ecdc.europa.eu/ under a CC BY license, with permission from European Center for Disease Prevention and Control, original copyright 2021.

Other statistical approaches for the early detection of outbreaks of infectious diseases are syndromic surveillance systems, which rely on health-related data collected electronically for other purposes. Among these alternative data sources are primary care physicians, emergency rooms, intensive care units, pharmacies, etc. These data are monitored to trigger an alarm indicating the possibility of an outbreak whenever deviations from the usual pattern are observed [7]. However, setting the control levels above which the system should signal the alarm still poses a major challenge.

Oehmke et al. [8] extended the traditional surveillance measures with indices aimed at informing on the rate of COVID-19 transmission. They measure speed, acceleration or deceleration, change in acceleration or deceleration (jerk), and 7-day transmission rate persistence. Following the methodology exposed in [9], the daily number of positive cases is a function of the prior number of cases, the level of testing, and other variables that measure the trend of the weekly contagion rate. The main goal is to compare COVID-19 transmission activity in the current week relative to previous weeks. The research concluded that the standard surveillance metrics are necessary but insufficient to mitigate and prevent COVID-19 transmission. These studies demonstrated the need to complement traditional surveillance indicators, such as the commonly used $I_{14}$ index, to describe the dynamics of the pandemic. In our research, we enhance this dynamic approach to pandemic surveillance by developing a methodology based on probabilistic and statistical data analysis that yields early warnings of the start of a new pandemic wave as soon as significant changes in the monitored surveillance index are detected. We define a new index, based on the daily new-case count, analyze its variability in a stationary situation, and determine the threshold values above which the pandemic activity shows a significant change of trend in the population. The variability model considers transmission within and outside households, resulting in the over-dispersion observed in the data [10]. In summary, the novelty of this research is the monitoring of a new dynamic surveillance index providing statistically-founded threshold values at which to set automatic alerts of a change of trend in the data. Thus, we obtain a more suitable pandemic monitoring tool for earlier warning of the onset of a new wave.

The rest of the paper is structured as follows: The Methods section introduces the weighted cumulative incidence (WAI) index and shows how it is monitored using control charts. Results section demonstrates the application of the proposed methodology with both synthetic and real data to point up the advantages of the WAI index with respect to the $I_{14}$ index when it comes to anticipating changes in the spread of the virus. Discussion section contains the discussion and conclusions.

## Methods

### The weighted cumulative incidence (WAI) index

Let $Y(t)$ be the new cases per 100,000 population detected on day $t$. As noted in the Introduction, the 14-day notification rate of new COVID-19 cases, $I_{14}(t) = \sum_{j=0}^{13} Y(t-j)$, is the main indicator used for monitoring the evolution of the pandemic over time. However, this value may be not representative of the current spread of the virus among the population, nor a good approximation of the rate at which positive cases change at time $t+1$. Note that the difference in its value between two consecutive days is simply the difference in positive cases between the day before and 14 days ago.

$$I_{14}(t+1) - I_{14}(t) = \sum_{j=0}^{13} Y(t+1-j) - \sum_{j=0}^{13} Y(t-j) = Y(t+1) - Y(t-13) \qquad (1)$$

We propose the use of a new index that is better able to reflect both the current state of

virus transmission at time $t$ and the current rate of change. With the new index, denoted by $Z_{14}(t)$, the last observations of the series $\{Y(t)\}$ are weighted exponentially:

$$Z_{14}(t) = \frac{\alpha}{1 - (1 - \alpha)^{14}} \sum_{j=0}^{13} (1 - \alpha)^j Y(t - j) \tag{2}$$

where $0 < \alpha \leq 1$ is a weighting coefficient that controls the weight decay over time. The new index $Z_{14}(t)$ is defined as a linear combination of the last 14 observations. Because a value other than 14 could be considered, throughout the following mathematical analysis, 14 is substituted by a general value P, and the index is denoted by $Z_P(t)$.

The weights of the linear combination $\frac{\alpha}{1 - (1 - \alpha)^P} (1 - \alpha)^j$ decrease geometrically with the age of the data and sum to unity, since

$$\sum_{j=0}^{P-1} (1 - \alpha)^j = \frac{1 - (1 - \alpha)^P}{\alpha} \tag{3}$$

## Monitoring the WAI index using control charts

The WAI index fluctuates around an average value $E(Y(t)) = \mu_0$, when the spread of the virus is under control. We define a Control Chart to monitor the WAI index in order to detect significant variations in the virus transmission process; that is, to detect statistically significant departures from the stable value $\mu_0$.

The limits of the Control Chart are set at $K$ times the standard deviation of $Z_P(t)$ from its expected value $\mu_0$. The variance of $Z_P(t)$, $V(Z_P(t))$, is calculated assuming a Poisson distribution for the number of new infections.

$$V(Z_P(t)) = \frac{\alpha^2}{(1 - (1 - \alpha)^P)^2} \sum_{j=0}^{P-1} (1 - \alpha)^{2j} \mu_0 = \left( \frac{\alpha}{2 - \alpha} \frac{1 - (1 - \alpha)^{2P}}{(1 - (1 - \alpha)^P)^2} \right) \mu_0 \tag{4}$$

And the lower and upper control limits, LCL and UCL, respectively, are

$$UCL = \mu_0 + K\sqrt{V(Z_P(t))} \tag{5}$$

$$LCL = \mu_0 - K\sqrt{V(Z_P(t))} \tag{6}$$

The value for parameter $K$ is set to three, in accordance with usual practice in process control.

## Virus transmission inside and outside households: A model for overdispersion in the observed data

In the Poisson distribution, expected variance equals the mean, $V(Y(t)) = E(Y(t))$. However, overdispersion, which exists when observed variance exceeds expected variance, has been observed in a daily new-case count series during a disease outbreak [11, 12]. We propose to model infection spread at two levels, inside and outside the home, to explain the overdispersion observed in the data. A household unit is defined as a group of people living together. We consider a household to be uninfected when none of its inhabitants is infected, otherwise, the household is classed as infected. The household unit becomes infected when one of its members is infected by an outside contact, in which case the remaining members have a high probability of becoming infected.

The daily new-case count is expressed as the following random sum

$$Y(t) = \sum_{i=1}^{F(t)} C_i \tag{7}$$

where, $F(t)$ is the number of infected households detected at time $t$ and $C_i$ is the number of individuals infected in household $i$, which is a random variable with mean $E(C) = \mu_C$ and variance $V(C) = \sigma_C^2$.

Assuming that $F(t)$ has a Poisson distribution, then Wald's equation provides the expected value of $Y(t)$:

$$E(Y(t)) = E(\sum_{i=1}^{F(t)} C_i) = \mu_{F(t)}\mu_C \tag{8}$$

The variance of Y(t) is (see Proposition 1 in S2 Appendix):

$$V(Y(t)) = \mu_{F(t)}E(C^2) \tag{9}$$

Because $C \geq 1$, then $E(C^2) > E(C)$ and therefore $V(Y(t)) > E(Y(t))$, and it can be concluded that variable $Y(t)$ is over-dispersed (see Corollary 1 in S2 Appendix).

Then, when the transmission of the virus is stable ($\mu_{F(t)} = \mu_F$), the mean and the variance of the WAI index $Z_P(t)$ are

$$E(Z_P(t)) = \mu_F\mu_C \tag{10}$$

$$V(Z_P(t)) = \left( \frac{\alpha}{2 - \alpha} \frac{1 - (1 - \alpha)^{2P}}{(1 - (1 - \alpha)^P)^2} \right) \mu_F E(C^2) \tag{11}$$

which are used in (5) and (6) to define the limits of the Control Chart for the $Z_P(t)$ index.

The expressions for $\mu_F$, $\mu_C$ and $E(C^2)$ depend on the ratio of new infections caught at home, denoted by a parameter $g$, and the household size distribution, denoted by the variable $H$ (see S2 Appendix). The value $g$, which is usually obtained from back tracing, is reported by public health administrations and $H$ is also reported in official statistics.

$$\mu_F = (1 - g)E(Y(t)) \tag{12}$$

$$\mu_C = \frac{1}{1 - g} \tag{13}$$

$$E(C^2) = \frac{\mu_H E(H^3)}{E^2(H^2)(1 - g)^2} \tag{14}$$

## Including the seasonal component

The data $Y(t)$ represent the normalized daily number of individuals testing positive on a SARS-CoV-2 diagnostic test. There are several tests with varying degrees of performance in terms of rapidity; sensitivity, specificity, (some fulfill both the latter criteria); and price. These characteristics affect the nature and quality of collected data for deriving an approximation of the real number of new infections in the population.

Based on the detection system used by the public health services, the daily new-case count exhibits seasonality arising from differences between working days and weekends (see Fig 2).

**Original data series**

Data from Andalucía

**Fig 2. Daily new-case count series for the Andalusian Region of Spain, showing the weekly seasonal component.**

This seasonal component needs to be incorporated into the statistical analysis and the observations deseasonalized following the method proposed in S1 Appendix.

We denote by $f_L$ the seasonal factor for non-Monday and non-post-holiday working days, $f_H$ the seasonal factor for weekends and holidays and $f_M$ the seasonal factor for Mondays and post-vacation working days. The deseasonalized data series, $X(t)$, is $X(t) = f_L^{-1} Y(t)$ when $t$ is a working day, $X(t) = f_H^{-1} Y(t)$ when $t$ is a weekend or holiday and $X(t) = f_M^{-1} Y(t)$ when $t$ is a post-vacation working day.

The expected value and variance of $X(t)$ depends on the type of day:

$$E(X(t)|t \in L) = f_L^{-1} \mu_{F(t)} \mu_C \quad V(X(t)|t \in L) = f_L^{-1} \mu_{F(t)} E(C^2)$$

$$E(X(t)|t \in H) = f_H^{-1} \mu_{F(t)} \mu_C \quad V(X(t)|t \in H) = f_H^{-1} \mu_{F(t)} E(C^2) \tag{15}$$

$$E(X(t)|t \in M) = f_M^{-1} \mu_{F(t)} \mu_C \quad V(X(t)|t \in M) = f_M^{-1} \mu_{F(t)} E(C^2)$$

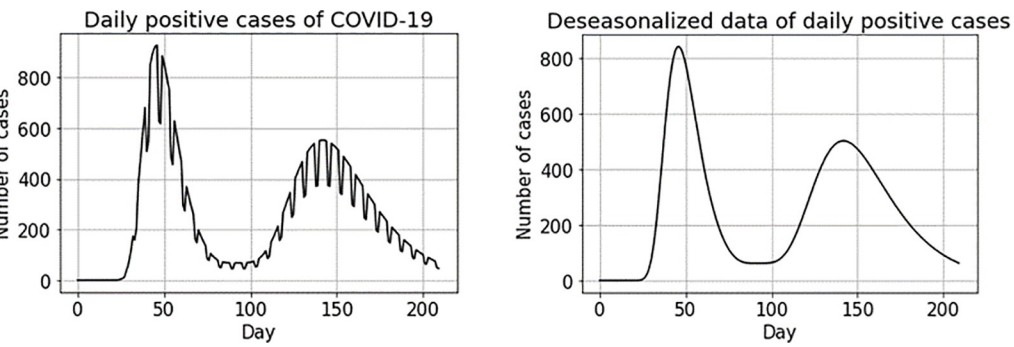

**Fig 3. Original data series (left) and deseasonalized data series (right) for two consecutive waves of the pandemic.**

Therefore, considering the seasonal factors, the variance of the index $Z_P(t)$ defining the control limits in expression (5) and (6) is

$$V(Z_P(t)) = \frac{\alpha^2}{(1 - (1-\alpha)^P)^2} \sum_{j=0}^{P-1} (1-\alpha)^{2j} \mu_{F(t)} E(C^2) \left( f_L^{-1} 1_{\{t \in L\}} + f_H^{-1} 1_{\{t \in H\}} + f_M^{-1} 1_{\{t \in M\}} \right) \quad (16)$$

## Results

This section begins with the application of the methodology with synthetic data in order to illustrate the advantages of the WAI index over the classic $I_{14}(t)$ index as a detection mechanism for changes in the spread of the virus. Then, the monitoring is applied to real data for different regions of Spain and different pandemic phases.

### Comparison of the WAI index with the 14-day cumulative new-case index

We compare the performance of the indices, $I_{14}(t)$ and $Z_{14}(t)$, for approximating the real virus transmission process. To represent both indices on the same scale as the data series, the $I_{14}(t)$ index is divided by 14. We use a mathematical model that mimics the real behavior of daily cases during and between pandemic waves.

The graph on the left in Fig 3 represents registered new cases, while the one on the right depicts the situation after deseasonalizing the data series.

Fig 4 depicts the new-case count series (in black), the values of the $I_{14}(t)$ index (red line) and the WAI index (green line). Both indices follow the shape of the original data series with

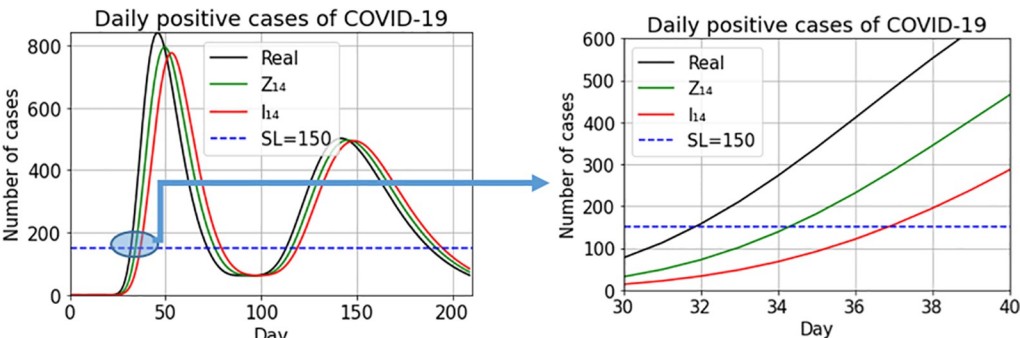

**Fig 4. Daily positive COVID-19 cases (black line), $I_{14}(t)$ index (red line) and WAI index (green line).**

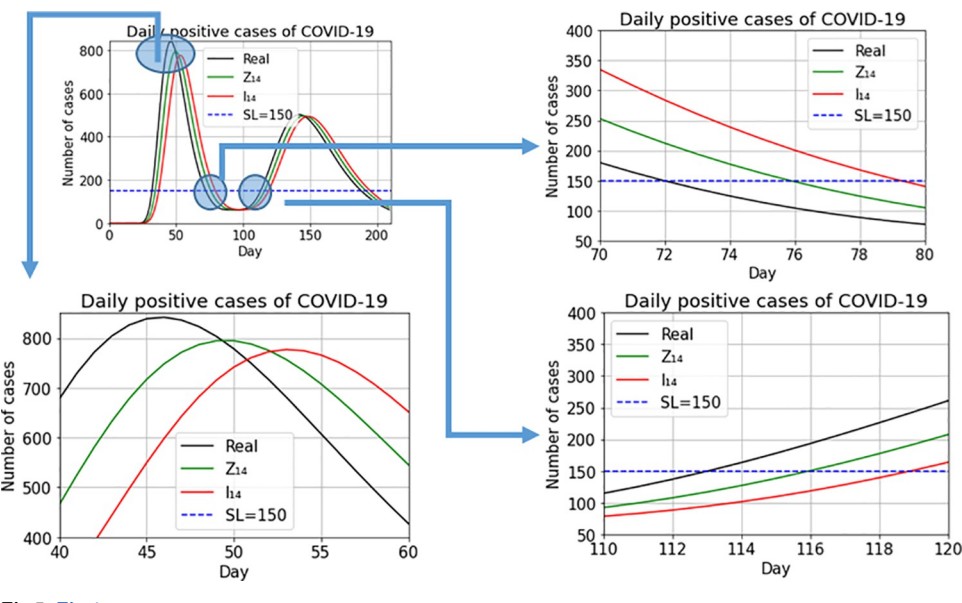

**Fig 5. Fig 4 zoom.**

the $I_{14}(t)$ index always lagging behind the WAI index. To assess the impact of this lag, we consider a commonly-accepted high public health risk threshold, shown with a dashed blue line. In crossing this threshold, the WAI index lags two days behind the real series and the normalized $I_{14}(t)$ index lags five days behind the real series. The right-hand graph in Fig 4 is a zoom of the area in which the two indices and the data series cross the alert threshold. By the time the normalized $I_{14}(t)$ index rings the alarm bell, the real series more than triples the risk threshold, while this difference with the WAI index is much smaller. Therefore, to detect changes in the virus spread pattern, it is important to use indices as close as possible to the real data series. Delay in detecting a change in pattern is important, not only at the onset of a wave, but also at other pandemic moments such as peaks and stabilization periods. Fig 5 zooms into the graph at other points in the trajectory of the pandemic: at the peak of the wave, the $I_{14}(t)$ index marks the onset of the decline in the cycle at 7 days past the peak; that is, 4 days later than the WAI index. Similar lags can be observed between the points at which the two indices dip below the high-risk threshold and between their points of upturn with the onset of this new wave.

## Monitoring new cases of COVID-19 for early alert

In this section, the methods presented above are used to monitor the series of positive cases as one wave fades away, in order to sound the alert for the onset of a new one. Real data for various regions of Spain are used to illustrate the methodology.

Fig 6 Left shows the original data series of positive cases recorded in the region of Navarre from July 1, 2020 to November 11, 2021. The last pandemic wave reached its peak on the 15[th] of July before subsiding then stabilizing around the 22[nd] of September. The seasonal components are calculated as $f_L^{-1} = 0.938, f_H^{-1} = 1.17, f_M^{-1} = 0.757$, and the data series is deseasonalized (see Fig 6 Right).

The mean daily number of cases is estimated by averaging the deseasonalized data for the period 22[nd] of September to 12[th] of October, $\mu_0 = 19.93$. This average value, together with the estimated ratio of household transmission reported by the public health administration [13], $g = 0.4$, provides the expected daily number of infected households $\mu_{F(t)} = 11.96$ and the expected

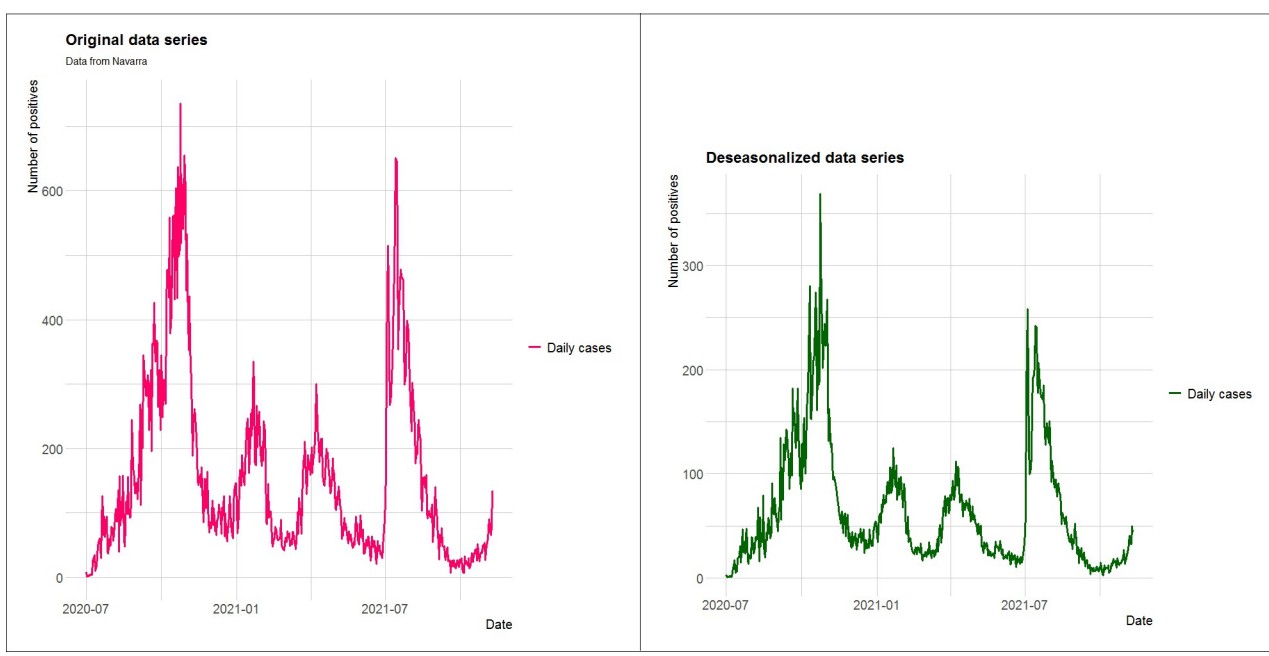

**Fig 6. Original new-case count series (Left) and Deseasonalized data series (Right) for the Spanish region of Navarre.**

number of infections per household $\mu_C$ = 1.67. In addition, using the household size distribution (see Fig 7) reported by the National Institute of Statistics [14] to estimate $E(C^2)$ = 3.32, the variance of the WAI index is calculated as:

$$V(Z_P(t)) = \frac{0.2^2}{(1 - (1 - 0.2)^P)^2} \sum_{j=0}^{P-1} (1 - 0.2)^{2j}(11.96)(3.32)\left[(0.938)1_{\{t \in L\}} + (1.17)1_{\{t \in H\}} + (0.757)1_{\{t \in M\}}\right] \quad (17)$$

The central value of the Control Chart is set at the mean value and the control limits at $UCL = 19.93 + 3\sqrt{V(Z_P(t))} \ and \ LCL = 19.93 - 3\sqrt{V(Z_P(t))}$.

Fig 8 shows the monitoring of pandemic data by the Control Chart and the WAI for the region of Navarre. The stable period used to estimate the statistics is indicated by dashed lines. It can be seen that virus transmission remained stable for four days, but a first out-of-control signal was returned on October 17[th], after which the WAI index remained out of control. We compare this alert system with the one provided by the $I_{14}(t)$ index. Fig 8 shows that the index remained stable until October 16[th], when an increasing trend resulted in the crossing of the first alert threshold (set at 100 cases per 100,000 population) on November 5[th], 18 days later than indicated by the WAI index.

Observe that the $I_{14}(t)$ index series signals an alarm when the variation in the spread of the pandemic surpasses a fixed threshold (red dot in Fig 8). This risk threshold is usually set high, at, say, 75 or 100 cases per 100,000 population. The proposed WAI index, however, adjusts to variability during the period of stability in the region under analysis. The Control Chart keeps the observation within the control limits (period of stability), with random variations, and the alarm for a change in virus transmission is triggered as soon as the control limit is surpassed (red square in Fig 8).

The behavior described for Navarre is also observed in other regions of Spain, as shown in Fig 9.

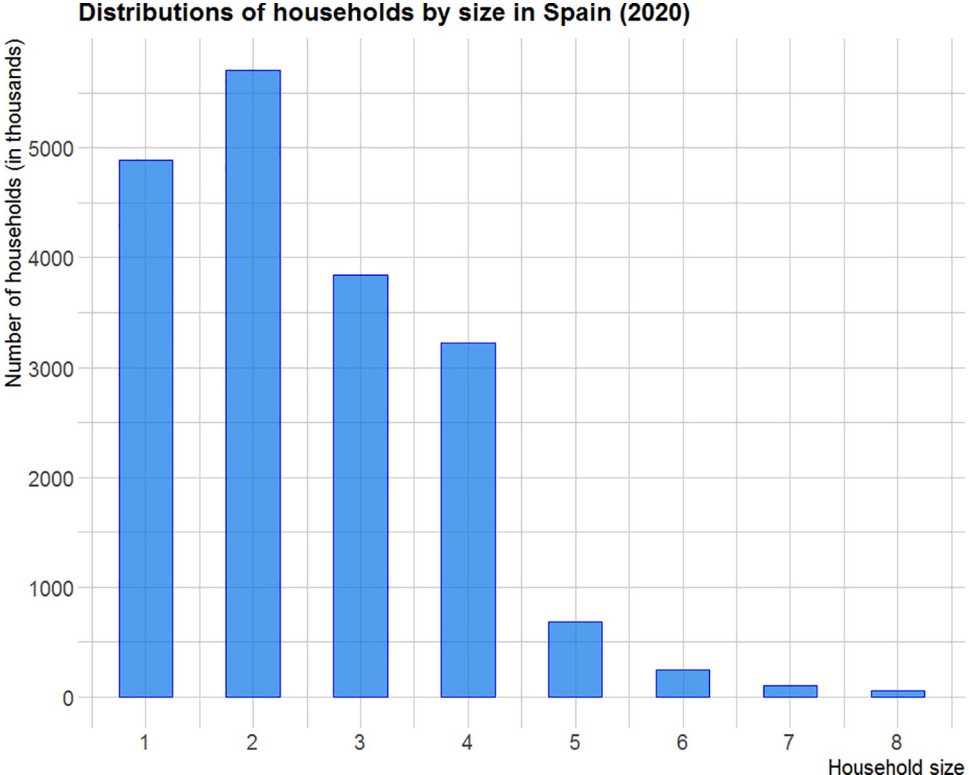

**Fig 7. Spanish household size distribution (source: National Institute of Statistic).**

## Discussion

### Principal findings

The WAI index $Z_{14}(t)$ is a statistically-validated new-case index for COVID-19 and a highly valuable addition to the set of epidemiological metrics for understanding the progression of the pandemic. We find that it reacts more quickly to changes in transmission activity than the standard surveillance metric $I_{14}(t)$ (14-day notification rate of reported COVID-19 cases per 100,000 population). The probabilistic analysis carried out for the variability of the $Z_{14}(t)$ index enables the construction of a Control Chart for use as a surveillance tool for triggering automatic alerts for significant variations in regional epidemic activity. We find that separate modeling of virus transmission inside and outside households explains the overdispersion observed in the data.

Time series charts have been used by health administrations to monitor COVID-19 pandemic indices such as numbers of infections, hospitalizations, ICU admissions and deaths. Without statistical support, time series charts are prone to over-interpretation by the public, the health authorities and political leaders, due to chance fluctuations in daily data [15], which can then influence health policy decisions and individual behavior. Governments decide to reopen after a lockdown based on a downturn in the trajectory of new cases or hospitalizations, or to reinstate distancing measures when detecting an upturn. We have developed a Control Chart which enables the distinction of chance from systematic variations in processes and their outcomes. Control Charts show whether the situation is deteriorating, remaining stable, or improving. In fact, statistical process control is a widely used statistical technique for monitoring health-related processes and identifying changes in them. We refer readers to

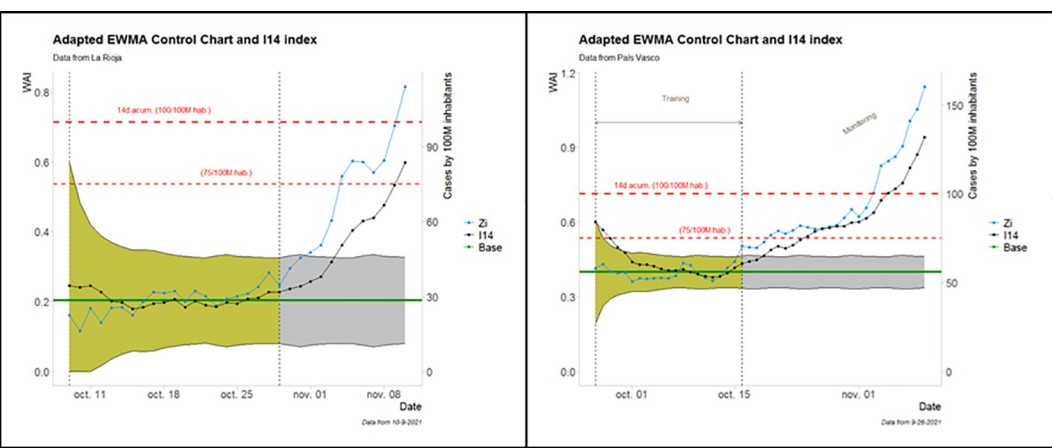

**Fig 8. Monitoring of pandemic data by the control chart and the WAI for the region of Navarre.** The blue line depicts the WAI value, the black line refers to the $I_{14}$ index and the green one is the mean value of $Z_i$ in the training period. The shaded band contains the admissible variability of the series and shows when to trigger the alert if a new observation falls outside the band. The yellow area is the model-training period and the gray area is the monitoring period. The red lines illustrate two typical reference levels for the $I_{14}$. The principal y-axis contains the WAI values, while the secondary y-axis contains the respective number of cases per 100M inhabitants.

[16, 17] and references therein for an extended discussion of the use of control charts in health care and public health surveillance. Control charts have been used to assess the effect of government interventions during the COVID-19 pandemic [18, 19]. However, the Control Chart

**Fig 9. Control Charts for regions of The Rioja (Left) and Basque Country (Right).**

for monitoring the $Z_{14}(t)$ index is the first attempt at using them as a surveillance tool in the COVID-19 pandemic. With respect to the monitoring of count data, control charts have been used for monitoring the number of adverse events in a pharmaceutical company [20] and for monitoring the annual incidence of male thyroid cancer [21]. In closer relation to our own application, monitoring has been also used to identify early warning signs in an ED in the face of an abnormally high patient influx [22]. This last reference, however, describes a different monitoring strategy combining autoregressive-moving-average (ARMA) time series models with the generalized likelihood ratio (GLR) chart. The use of complex time series models requires a sample size that may not be available when dealing with the novel and rapidly changing phenomena associated with pandemics. Sahai et al. [23] used data from February 15, 2020 to June 30, 2020 to fit Auto Regressive Integrated Moving Average (ARIMA) models to predict the incidence and spread of the COVID-19 in five countries (model specification was India (4,2,4), Brazil (3,1,2), Russia (3,0,0), Spain (4,2,4) and US (1,2,1)). They pointed out that the ARIMA forecast, built on the autoregressive nature of the time series coupled with corrective incremental adjustments, essentially predicts a linear pattern and fails to predict a series with turning points. In contrast with this approach, the goal of our research is not to forecast the future trajectory of the pandemic, but to ensure its earliest possible detection, by using constantly updated real-time data, while the incidence of the disease is changing, especially in an upward direction, in anticipation of a new pandemic wave.

We find that the WAI index is able to detect significant changes in regional epidemic activity more quickly than the commonly-used cumulative incidence ratio $I_{14}(t)$. The monitoring of $Z_{14}(t)$ index using a Control Chart enables alert signals to be triggered if a new observation trespasses the control limits. When this happens, the observation should be checked for recording errors. If the value is validated, then the data do not meet the assumption of stability in the spread of the virus, which may signal the start of a change in the transmission dynamics.

## Rationale of the probabilistic model for the $Z_{14}(t)$ index

The $Z_{14}(t)$ index monitoring system is inspired by the EWMA control charts, which are more suitable for the early detection of shifts in the parameters under surveillance than Shewhart-type attribute control charts, because, being equipped with memory, they are able to incorporate information from past observations and thus enable more rapid detection of variation in the mean of the process [24].

Transmission of the virus by infected individuals is the result of their viral shedding combined with their contact pattern [25]. Both these factors vary among infected individuals, making the transmission of the virus a stochastic process. Reports indicate an increased risk of infection following prolonged exposure in a confined space with at least one infected individual [26]. The fact that these are the prevailing conditions in a household with one infected member is what provided the motivation for the development of a clustering model of transmission process dynamics that differentiates between infection inside and outside the home [27]. The percentage of home infections is frequently picked up and reported by public health administrations through back tracing. In addition, the cluster-based probabilistic model uses the Poisson distribution to model random outside infections. Infections that occur outside the home exhibit a more random pattern due to unforeseen contacts with infected individuals. The transmission of the virus from one person to another through this secondary route on a specific day can be regarded as a low-probability event for each individual and, in the case of a large population living apart without contact, an independent event among the majority of individuals. Therefore, it is reasonable to model such transmission using a Poisson random variable. The Poisson distribution is a discrete probability distribution commonly used for

counting random events occurring in a given interval of time [28]. A great number of studies (see for example, [29–32]) use a Poisson Process to model patient arrivals at healthcare centers. The Poisson model has also been used for virus transmission modeling during this pandemic [33]. It is sometimes used in combination with other distributions, as in [34], where the rate of the Poisson distribution is assumed to follow a Gamma distribution, resulting in a negative binomial distribution. These models rely on differential equation models based on population dynamics [35, 36]. One of the most widely-used models of human-to-human transmission is the SIR model [37]. Extensions of the classical SIR model have indeed been developed for the COVID-19 pandemic [38–44]. However, such models are based on a set of adaptive parameters whose values only become available when the pandemic has passed. The approach presented in this paper is much simpler, while also effective for monitoring the spread of the virus. The household-cluster probabilistic (HCP) model counts the daily number of *infection-outside-the-home* events using the Poisson distribution; while infections inside the home are modeled by a random variable whose statistical properties depend on household size (see S2 Appendix). The resulting model is a parsimonious data-driven model whose parameters are easy to obtain from ordinary official reports.

To apply Wald's Eq in (8), it is necessary to assume the independence among the terms of the summation (and their identical distribution) as well as the independence of these terms with their count. Both assumptions can be considered to hold if we assume that the number of infected individuals within each household results from an independent internal transmission process, regardless of what happens within each of the other infected households or the total number of infected households. While there may be situations where this assumption does not hold, it appears to be a reasonable assumption. Assuming these hypotheses, the variance of the Z14 index is computed in S2 Appendix.

## Limitations

One limitation of the proposed methodology lies in the difficulty of applying it in the context of an outbreak caused by a novel virus. This arises due to the requirement of obtaining data related to the virus's transmission dynamics under conditions of stability, which may not be attainable during the initial stages of an outbreak or with rapidly evolving pathogens. In addition, the method is specifically designed to identify and alert for new outbreaks, but it does not provide information regarding the conclusion of a pandemic. This method does not consider undercounts, biased or missing data, and may be limited in cases of suboptimal public health surveillance systems. The reliance on available data introduces the possibility of incomplete or inaccurate reporting, impacting the reliability of the results. Robust surveillance systems and careful consideration of data quality are crucial when implementing this methodology.

This study uses the data collected by the Spanish Ministry of Health [45], which are defined as follows: COVID-19 cases are reported to the National Epidemiological Surveillance Network (RENAVE) through the web-based SiViES (Spain's Surveillance System), managed by the National Center for Epidemiology (CNE). This information comes from the case epidemiological survey that each Autonomous Community completes upon identifying a COVID-19 case. The imputed date corresponds to the symptom onset date or, if not available, the diagnosis date minus 6 days (from the beginning of the pandemic until May 10, 2020), minus 3 days (from May 11, 2020, until March 28, 2022), and minus 2 days from March 28, 2022 onwards. For asymptomatic cases, the diagnosis date is used.

When data are not reported instantaneously or multiple-day data are aggregated into a single value, a non-normal zero value series is followed by a sudden spike in the data. Our analysis partially addresses this issue by using weighted averages in the definition of the $Z_{14}(t)$ index.

When there is a regular pattern in the collected data, these problems can also be partially overcome by deseasonalizing the time series. When data are poor, the Control Chart of $Z_{14}(t)$ index could generate false alerts and overestimation of the variance of the data series. Another potential limitation would be a lack of data on the household size distribution, which, in developed countries, is usually drawn from the population census. In addition, our estimation of the size distribution of infected households in Proposition 3 (S2 Appendix) assumes an infection probability proportional to household size. While we consider this assumption reasonable, it lacks empirical support, potentially resulting in a distribution that deviates from real-world data. If available, utilizing actual observed data would enhance the accuracy of the distribution.

## Validation and real use of the methodology

The numerical analysis presented in the Results Section shows the power of the Control Chart to keep the observation within the control limits when there is no substantial change in the new-case data. Thereby it enables the identification of upward and downward movements in daily data as random variations, while up-crossing of the limit can be interpreted as a significant change in the disease incidence, presaging the beginning of a new wave.

The WAI index and the Control Chart were implemented in a computational tool developed in R (see S3 Appendix) and have been used by the Government of the Spanish Autonomous Region of Navarre for the surveillance of virus propagation trends following the waves of infection experienced to date. After the third wave (February 2020), moreover, a Control Chart for each Spanish region was submitted to the Spanish Ministry of Health. Control Charts were also implemented for smaller geographical areas, such as municipalities. Automated monitoring generated a daily report showing the areas whose Control Charts had issued an alert. The practical use of the Control Charts confirmed the role of constant surveillance and the updating of preparedness and response plans after each wave of the pandemic.

Observe that data may experience a delay in notification, although not excessively long in our case, with most delays ranging between 0 and two days. This delay, inherent to the surveillance and data collection system, is a consistent factor throughout the entire data series and can lead to underestimations in the most recent data. Consequently, both the Z(14) value and the I14 index may underestimate the true incidence rate. However, since the index is designed to promptly detect changes in trends, and the control chart's central and upper limits depend on the observed values in the series, the small, steady bias in underestimating the incidence does not hinder the detection of trend changes using the control chart.

## Conclusions

Surveillance systems require a variety of metrics. The strength of this study is the derived new COVID-19 transmission metric and the Control Chart for its monitoring. One of the distinct characteristics of a pandemic is the need for very early detection of a new outbreak, because a short initial period of steady rise in the number of new cases is sometimes followed by one of exponential growth. Thus, surveillance systems, such as the one presented in this study for monitoring and controlling the spread of the pandemic, which trigger the alert as soon as a change in epidemic activity occurs, provide valuable support for informed health policies.

## Supporting information

**S1 Appendix. Steps in the construction of the deseasonalized data series X(t) and calculation of the seasonal factors $f_L$, $f_H$, y $f_M$.**
(PDF)

**S2 Appendix. Mathematical proofs for the Household Cluster Probabilistic (HCP) model.**
(PDF)

**S3 Appendix. R-code.**
(PDF)

## Author Contributions

**Conceptualization:** Marta Cildoz, Martin Gaston, Fermin Mallor.

**Data curation:** Marta Cildoz, Laura Frias, Daniel Garcia-Vicuña.

**Formal analysis:** Marta Cildoz, Martin Gaston, Laura Frias, Daniel Garcia-Vicuña.

**Funding acquisition:** Martin Gaston, Fermin Mallor.

**Investigation:** Marta Cildoz.

**Methodology:** Marta Cildoz, Martin Gaston, Laura Frias, Daniel Garcia-Vicuña, Cristina Azcarate, Fermin Mallor.

**Project administration:** Fermin Mallor.

**Software:** Martin Gaston, Laura Frias.

**Supervision:** Cristina Azcarate, Fermin Mallor.

**Validation:** Marta Cildoz, Laura Frias, Daniel Garcia-Vicuña, Fermin Mallor.

**Writing – original draft:** Cristina Azcarate, Fermin Mallor.

**Writing – review & editing:** Marta Cildoz, Martin Gaston, Laura Frias, Daniel Garcia-Vicuña, Cristina Azcarate, Fermin Mallor.

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
