## [Decision Letter · Decision Letter 0]

4 Jul 2023

PONE-D-23-10336Early detection of new pandemic waves. Control Chart and a new Surveillance Index.PLOS ONE

Dear Dr. Mallor,

Thank you for submitting your manuscript to PLOS ONE. After careful consideration, we feel that it has merit but does not fully meet PLOS ONE’s publication criteria as it currently stands. Therefore, we invite you to submit a revised version of the manuscript that addresses the points raised during the review process. In the discussion section please clearly address the limitations and their consequences. Take a careful look at the major comments of the reviewers.

We look forward to receiving your revised manuscript.

Kind regards,

Ralf Reintjes, PhD, MD, MSc(P.H.), MSc(Epi.)

Academic Editor

PLOS ONE

“All the authors acknowledge the support by grant PID2020-114031RB-I00 (AEI, FEDER EU) and SISCOVID (Ref: 0011-3638-2020-000006).”

Reviewers' comments:

Reviewer's Responses to Questions

**Comments to the Author**

1. Is the manuscript technically sound, and do the data support the conclusions?

Reviewer #1: Yes

Reviewer #2: Yes

2. Has the statistical analysis been performed appropriately and rigorously? 

Reviewer #1: Yes

Reviewer #2: Yes

3. Have the authors made all data underlying the findings in their manuscript fully available?

Reviewer #1: Yes

Reviewer #2: Yes

4. Is the manuscript presented in an intelligible fashion and written in standard English?

Reviewer #1: Yes

Reviewer #2: Yes

5. Review Comments to the Author

Reviewer #1: There are some limitations in general that are not flaws, just limitations that merit mentioning. 1. This method requires a WAI Index and controlled charts to compare alarming growth. While the methods are sound, a limitation is this cannot be used in the face of a novel virus, flu, etc that results in an outbreak. This is a solid methodology to use for an already known virus demonstrating exponential growth. 2. This method is informative of new outbreaks but less informative of the end of the pandemic. 3. This method outperforms standard surveillance as an epidemic expands. 4. This method does not control for undercounts and biased or missing data.

Reviewer #2: The manuscript entitled "Early detection of new pandemic waves. Control Chart and a new Surveillance Index." presents a new index to monitor an epidemic. The proposed index was already implemented to monitor COVID-19 cases in Navarro, Spain. The manuscript is well written, the proposed index is clever and has a valid motivation. I read the manuscript with great interest and I did some critics/commentaries regarding the methods and results. In general it seems to be a good alternative to I14 index used in Europe.

Major comments:

- Methods, page 5. Definition of Y(t). New cases per 100k inhabitants detected on day t. It is not clear if are cases reported on day t (scenario 1) or cases that occur on day t (scenario 2). There is a difference between these two scenarios. In scenario 1, a case that occur a month ago but just now the PCR result is know could be reported and counted today on day t. On the other hand, if the definition on scenario 2 is used then the know case would be counted a month ago, so Y(t-"1 month") would be added with another unit. The second scenario is better to describe disease dynamic but there is notification and laboratory delay to be dealt with.

- Page 6, equation 2. The Z14 index is very clever, I like its simplicity because it does make more sense to give more weight to more recent observations. However it is important to make clear the definition of Y(t) because the most recent observation may suffer from notification delay and the most recent cases may be underestimated.

- Z14 variance, page 5, equation 4. Independence among {Y(t),Y(t-1),...,Y(t-P+1)} is a very strong assumption, Y(t) are cases of infectious disease therefore they are not independent by definition. The number of cases on day t, wouldn't be so different then the number of cases on days t-1 or t+1, since some cases could be related, a family outbreak for instance. Having said that, my feeling is that adding the temporal dependence would reduce the variance of Z14, hence the variance calculated on equation 4 could be seen as an upper bound.

- Page 7, lines 189-196. A similar argument regarding dependence of a highly transmissible infection disease could be applied on equations 8 and 9. Because the Wald's equation assume independence among Cs. However, adding the household component somehow incorporates possible household outbreaks. In both cases, the independence assumption may be a very strong assumption and should at least be mentioned in Methods and be discussed at the end of the manuscript.

- Page 8, lines 199-210. The household size, H, distribution. For COVID-19 it may be OK to assume that the household size of infected cases distribution is similar to the household distribution of the general population. Although it could be different if an intervention is applied. For example, if schools are closed it is less likely to see outbreaks in household with kids then the distribution of the household size of the general population may be different than the H distribution for the infected cases.

- Page 9, How are f_L, f_H and f_M estimated? For example, on page 11 line 285, how those values were obtained?

- Section 3.2. Why alpha = 0.2 was chosen? (equation 17)

- Page 12. I like the idea of using a stable period to define the thresholds for an early warning. Can a stable period be defined? In COVID-19 I can see you are using the period in-between waves, probably related to variants. Could you join different periods to estimated the thresholds?

Minor comments:

- Fig 1, at least the version I downloaded is hard to read.

- Page 4, lines 100-103. The authors should also mention the vaccination as another potential bias to seroprevalence studies.

- Page 9, I assume that L, H and M are mutually exclusive. So the definition of L (working days) should be adapted accordingly. L are the working days excluding Mondays and post-vacation working days.

- Fig 6 and 7, could be together.

- Fig 9 and 10 could be together as well.

6. PLOS authors have the option to publish the peer review history of their article (what does this mean?). If published, this will include your full peer review and any attached files.

Reviewer #1: No

Reviewer #2: **Yes: **Leonardo S Bastos

---

## [Author Response · Author response to Decision Letter 0]

15 Aug 2023

Dear Prof. Ralf Reintjes, 

We have sent a new version of the paper titled “Early detection of new pandemic waves. Control Chart and a new Surveillance Index.” by Cildoz, Gastón, Garcia-Vicuña, Frias, Azcarate, and Mallor (PONE-D-23-10336) for your review. This version has undergone a substantial revision, taking into account all comments of the two reviewers and the editor.

The following list is a summary of the main changes that have been introduced within the paper in the fulfilment of the requirements given by the Reviewers and Editor: 

 We have obtained copyright permission and improved Figure 1. 

 The Subsection 4.4 has been expanded to incorporate and further discuss the limitations identified by the reviewers.

 More comprehensive details have been provided regarding the interpretation and acquisition of the data by the health administration.

 The assumption of independence among variables required for applying probabilistic results, such as calculating the variance of the new index and utilizing Wald's equation, has been thoroughly addressed.

 Old Appendix A (new S1 Appendix) has been expanded to provide enhanced clarity on the methodology for deseasonalizing the data series and calculating the seasonal components.

 To enhance readability, the order of subsections 4.3 and 4.4 has been exchanged.

 We have diligently addressed all technical issues raised by the reviewers.

The detailed list of changes and our responses to the Reviewers and Editor are given point-to-point after this letter. We thank the Reviewers for their time spent reviewing our work. We hope that this version of the paper fulfills the Editor requirements and is suitable for publication within PLOS ONE journal.

Sincerely Yours,

F. Mallor

RESPONSE TO THE EDITOR’S COMMENTS

Early detection of new pandemic waves. Control Chart and a new Surveillance Index. 

PONE-D-23-10336

The new version of the paper follows the required style template. 

“All the authors acknowledge the support by grant PID2020-114031RB-I00 (AEI, FEDER EU) and SISCOVID (Ref: 0011-3638-2020-000006).” Please state what role the funders took in the study. If the funders had no role, please state: "The funders had no role in study design, data collection and analysis, decision to publish, or preparation of the manuscript." If this statement is not correct you must amend it as needed.

The funders had no role, so we have added the suggested sentence in the cover letter: "The funders had no role in study design, data collection and analysis, decision to publish, or preparation of the manuscript." 

 You may seek permission from the original copyright holder of Figure 1 to publish the content specifically under the CC BY 4.0 license. 

We have obtained written permission from the copyright holder (ECDC) under the CC BY 4.0 license to publish figure 1. 

We also have contacted with EuroGeographics, UN-FAO, Turkstat, Kartverket and Instituto Nacional de Estadística-Statistics Portugal. All of them gave as permission in different ways:

 EuroGeographics and Kartverket also sent a signed permission to publish content under CC-BY license.

 UN-FAO sent an email telling us that we only needed permission from ECDC.

 Turkstat and Instituto Nacional de Estadística-Statistics Portugal sent an email giving granted permission if the information is referenced, as indicated in their web pages: https://www.ine.es/dyngs/AYU/en/es/en/es/index.htm?cid=125 (INE) and https://www.tuik.gov.tr/Kurumsal/Yasal_Uyari (Turkstat)

All these documents and emails are uploaded to PLOS ONE as an "Other" file with our Revised submission.

Finally, In the figure caption of the copyrighted figure 1, we have included the following text: Reprinted from https://www.ecdc.europa.eu/ under a CC BY license, with permission from European Center for Disease Prevention and Control, original copyright 2021.

AppendixA, AppendixB and AppendixC have been removed from the main manuscript, and included as Supporting information files. Following the journal guidelines, a Supporting Information section has been included in the manuscript and the in-text citations have been updated.

 

RESPONSE TO THE REVIEWER #1

Early detection of new pandemic waves. Control Chart and a new Surveillance Index. 

PONE-D-23-10336

Reviewer: There are some limitations in general that are not flaws, just limitations that merit mentioning. 1. This method requires a WAI Index and controlled charts to compare alarming growth. While the methods are sound, a limitation is this cannot be used in the face of a novel virus, flu, etc that results in an outbreak. This is a solid methodology to use for an already known virus demonstrating exponential growth. 2. This method is informative of new outbreaks but less informative of the end of the pandemic. 3. This method outperforms standard surveillance as an epidemic expands. 4. This method does not control for undercounts and biased or missing data.

We thank to the reviewer for providing valuable and constructive comments regarding the limitations of the proposed method. We greatly appreciate their insights, which have prompted us to consider and address these limitations in our revised manuscript. In response to the reviewer's input, we have incorporated their points into the Discussion section, specifically in Subsection Limitations. The revised version of the paper now includes the following text:

One limitation of the proposed methodology lies in the difficulty of applying it in the context of an outbreak caused by a novel virus. This arises due to the requirement of obtaining data related to the virus's transmission dynamics under conditions of stability, which may not be attainable during the initial stages of an outbreak or with rapidly evolving pathogens. In addition, the method is specifically designed to identify and alert for new outbreaks, but it does not provide information regarding the conclusion of a pandemic. This method does not consider undercounts, biased or missing data, and may be limited in cases of suboptimal public health surveillance systems. The reliance on available data introduces the possibility of incomplete or inaccurate reporting, impacting the reliability of the results. Robust surveillance systems and careful consideration of data quality are crucial when implementing this methodology. 

RESPONSE TO THE REVIEWER #2

Early detection of new pandemic waves. Control Chart and a new Surveillance Index. 

PONE-D-23-10336

We thank the Reviewer for dedicating valuable time to thoroughly review our manuscript. Their insightful comments on the methods and results have greatly contributed to enhancing the clarity and presentation of our research. We are particularly grateful for their positive assessment, highlighting that the new index 'seems to be a good alternative to I14 used in Europe.' We believe that it can be a valuable complementary measure. 

Below is our answer point by point to your comments:

- Methods, page 5. Definition of Y(t). New cases per 100k inhabitants detected on day t. It is not clear if are cases reported on day t (scenario 1) or cases that occur on day t (scenario 2). There is a difference between these two scenarios. In scenario 1, a case that occur a month ago but just now the PCR result is known could be reported and counted today on day t. On the other hand, if the definition on scenario 2 is used then the know case would be counted a month ago, so Y(t-"1 month") would be added with another unit. The second scenario is better to describe disease dynamic but there is notification and laboratory delay to be dealt with.

Thank you for raising this important issue regarding the accurate definition of the data. We utilize the official data provided by the Spanish Ministry of Health, which can be accessed through their website (https://cnecovid.isciii.es/covid19/). I have included below the excerpt from the official webpage, which outlines how the data is obtained and its meaning. Based on the procedure explained for assigning a date to each case, the data align with your described scenario 2, although any delays between the new case date and notification date are typically only a matter of a few days rather than weeks or months. Naturally, the validity of conclusions drawn from any data analysis heavily relies on the quality of the underlying data. We employ the same raw data that health authorities utilize to construct the I14 index, which informs their decision-making process.

Taking into account your comment, as well as that of the other reviewer, we have expanded subsection regarding limitations to include an explanation of the data's meaning and the limitations associated with an imperfect surveillance and data collection system. The new text included is:

The reliance on available data introduces the possibility of incomplete or inaccurate reporting, impacting the reliability of the results. Robust surveillance systems and careful consideration of data quality are crucial when implementing this methodology. 

This study uses the data collected by the Spanish Ministry of Health [15] which are defined as follows: COVID-19 cases are reported to the National Epidemiological Surveillance Network (RENAVE) through the web-based SiViES (Spain's Surveillance System), managed by the National Center for Epidemiology (CNE). This information comes from the case epidemiological survey that each Autonomous Community completes upon identifying a COVID-19 case. The imputed date corresponds to the symptom onset date or, if not available, the diagnosis date minus 6 days (from the beginning of the pandemic until May 10, 2020), minus 3 days (from May 11, 2020, until March 28, 2022), and minus 2 days from March 28, 2022 onwards. For asymptomatic cases, the diagnosis date is used.

- Page 6, equation 2. The Z14 index is very clever, I like its simplicity because it does make more sense to give more weight to more recent observations. However it is important to make clear the definition of Y(t) because the most recent observation may suffer from notification delay and the most recent cases may be underestimated.

We appreciate the reviewer's observation, which aligns with the definition of the data presented and discussed in their previous comment. It is true that the data may experience a delay in notification, although not excessively long in our case, with most delays ranging between 0 and two days. This delay, inherent to the surveillance and data collection system, is a consistent factor throughout the entire data series and can lead to underestimations in the most recent data. Consequently, both the Z(14) value and the I14 index may underestimate the true incidence rate. However, since the index is designed to promptly detect changes in trends, and the control chart's central and upper limits depend on the observed values in the series, the small, steady bias in underestimating the incidence does not hinder the detection of trend changes using the control chart. Certainly, this is an important observation that we have included in the Validation and real use of the methodology subsection. The new text included is the previous one in red.

- Z14 variance, page 5, equation 4. Independence among {Y(t),Y(t-1),...,Y(t-P+1)} is a very strong assumption, Y(t) are cases of infectious disease therefore they are not independent by definition. The number of cases on day t, wouldn't be so different then the number of cases on days t-1 or t+1, since some cases could be related, a family outbreak for instance. Having said that, my feeling is that adding the temporal dependence would reduce the variance of Z14, hence the variance calculated on equation 4 could be seen as an upper bound.

The reviewer's analysis is correct, and in fact, it is this kind of reasoning that motivated the subsequent introduction of the household factor to differentiate between infections occurring inside and outside the home. Infections that occur within the household are highly dependent as they are practically unavoidable due to close cohabitation in the majority of cases. On the other hand, infections that occur outside the home follow a more random pattern due to unforeseen contacts with infected individuals. The transmission of the virus from one person to another on a specific day through this second route can be considered as a low-probability and independent event among the majority of individuals in a large population, and thus, can be reasonably modeled using a Poisson random variable. The discussion on this topic, included in Discussion Section, has been extended (text in red) to consider your comment in Subsection Rationale of the probabilistic model for the Z14 index, which states: Reports indicate an increased risk of infection following prolonged exposure in a confined space with at least one infected individual [27]. The fact that these are the prevailing conditions in a household with one infected member is what provided the motivation for the development of a clustering model of transmission process dynamics that differentiates between infection inside and outside the home [28]. The percentage of home infections is frequently picked up and reported by public health administrations through back tracing. In addition, the cluster-based probabilistic model uses the Poisson distribution to model random outside infections. Infections that occur outside the home exhibit a more random pattern due to unforeseen contacts with infected individuals. The transmission of the virus from one person to another through this secondary route on a specific day can be regarded as a low-probability event for each individual and, in the case of a large population living apart without contact, an independent event among the majority of individuals. Therefore, it is reasonable to model such transmission using a Poisson random variable. The Poisson distribution is a discrete probability distribution commonly used for counting random events occurring in a given interval of time [29]. A great number of studies (see for example, [30–33]) use a Poisson Process to model patient arrivals at healthcare centers. The Poisson model has also been used for virus transmission modeling during this pandemic [34].

On the other hand, if positive dependence were a predominant factor, as the reviewer rightly points out, the expression of variance would indeed be an upper bound. However, the observed phenomenon in the data (see references [10], [11], and [12]) is the opposite, that of overdispersion in the data. This greater variance in the data can be explained through the proposed household cluster model of infections as demonstrated in old Appendix B (new S2 Appendix), Mathematical proofs for the Household Cluster Probabilistic (HCP) model.

- Page 7, lines 189-196. A similar argument regarding dependence of a highly transmissible infection disease could be applied on equations 8 and 9. Because the Wald's equation assume independence among Cs. However, adding the household component somehow incorporates possible household outbreaks. In both cases, the independence assumption may be a very strong assumption and should at least be mentioned in Methods and be discussed at the end of the manuscript.

Indeed, to apply Wald's equation in (8), it is necessary to assume the independence among the terms of the summation (and their identical distribution) as well as the independence of these terms with their count. Both assumptions can be considered to hold if we assume that the number of infected individuals within each household results from an independent internal transmission process, regardless of what happens within each of the other infected households or the total number of infected households. While there may be situations where this assumption does not hold, it appears to be a reasonable assumption. Assuming these hypotheses, the variance of the Z14 index is computed in S2 Appendix.

As you suggest in your comment, we have added the previous paragraph (text in red) at the end of the Subsection Rationale of the probabilistic model for the Z14 index.

- Page 8, lines 199-210. The household size, H, distribution. For COVID-19 it may be OK to assume that the household size of infected cases distribution is similar to the household distribution of the general population. Although it could be different if an intervention is applied. For example, if schools are closed it is less likely to see outbreaks in household with kids then the distribution of the household size of the general population may be different than the H distribution for the infected cases.

We appreciate the reviewer's observation regarding the probability distribution of the size of infected households during the COVID pandemic. In old Appendix B (new S2 Appendix), Proposition 3 calculates the probability distribution of the household size variable for infected households by assuming that the probability of infection is proportional to the size of the household. For example, a household with four people has twice the probability of becoming infected compared to a household with a size of two. While this assumption of linearity for the infection probability is not supported by empirical data, we deemed it to be a reasonable approximation. Nevertheless, alternative assumptions about the probability of a household becoming infected as a function of its size could be considered in the calculations of Proposition 3, although this would lead to more intricate results. We have reflected this discussion in the subsection Limitations by adding the following text: Our estimation of the size distribution of infected households in Proposition 3 (S2 Appendix) assumes an infection probability proportional to household size. While we consider this assumption reasonable, it lacks empirical support, potentially resulting in a distribution that deviates from real-world data. If available, utilizing actual observed data would enhance the accuracy of the distribution.

- Page 9, How are f_L, f_H and f_M estimated? For example, on page 11 line 285, how those values were obtained?

The method for deseasonalizing the data series and calculating the seasonal components f_L, f_H, and f_M was explained in old Appendix A (new S1 Appendix) of the article. However, upon reviewing your comment, we have realized that the method was presented in a concise manner. In this revised version of the article, we have provided a more detailed explanation of the deseasonalization method in new S1 Appendix. The newly added text is highlighted in red:

S1 Appendix. Steps in the construction of the deseasonalized data series X(t) and calculation of the seasonal factors f_L, f_H y f_M

We have implemented the following deseasonalisation method included in [46].

 The series Y ®_7 (t) of moving averages of the original data series Y(t),t=1,…,n is calculated using seven terms, assuming a one-week period of oscillation. 

Y ®_7 (t)=(∑_(j=-3)^(+3)▒〖Y(t+j) 〗)⁄7 ∀t=4,…,n-3

 The original data Y(t) are divided by the results from step 1, to obtain the daily factors, distinguishing among three types of days: non-Monday, non-post-holiday working days (set L), holiday and weekend days (set H) and Mondays and post-holiday days (set M).

f_L (t)=Y(t)⁄(Y ®_7 (t) ) ∀ t∈L

f_H (t)=Y(t)⁄(Y ®_7 (t) ) ∀ t∈H

f_M (t)=Y(t)⁄(Y ®_7 (t) ) ∀ t∈M

 The average of seasonal factors is computed to obtain the seasonal components

f_L=(∑_(t∈L)▒〖f_L (t) 〗)⁄(#L) 

f_H=(∑_(t∈H)▒〖f_H (t) 〗)⁄(#H)

f_M=(∑_(t∈M)▒〖f_M (t) 〗)⁄(#M)

 The original data series Y(t) is divided by the (adjusted) seasonal factors to obtain the series X(t) of deseasonalized data.

X(t)=(Y(t))⁄f_D D=L ,H,M for t∈L,H,M,respectively

- Section 3.2. Why alpha = 0.2 was chosen? (equation 17)

This value was chosen based on a recommended parameter setting for the Exponentially Weighted Moving Average (EWMA) control chart. A discussion on the selection of the alpha value can be found in the classic book on quality control by Montgomery (Statistical Quality Control, 6th edition, Chapter 9, p. 423): “In general, we have found that values of alpha in the interval (0.05, 0.25) work well in practice, with alpha =0.05, alpha =0.10, and alpha =0.20 being popular choices. A good rule of thumb is to use smaller values of alpha to detect smaller shifts.”

- Page 12. I like the idea of using a stable period to define the thresholds for an early warning. Can a stable period be defined? In COVID-19 I can see you are using the period in-between waves, probably related to variants. Could you join different periods to estimate the thresholds?

We define a stable period as a time interval in which the daily series of accumulated incidence data does not exhibit a trend. In our application, this corresponds to the periods between pandemic waves. The question of whether it is possible to combine different stable periods is an interesting one. In general, the answer would be negative. This can be observed, for example, in the data from Navarra (Figure 5), where the stabilization levels of the series differ between different waves, while in the case of Andalucía (Figure 2), they are very similar.

Minor comments:

- Fig 1, at least the version I downloaded is hard to read.

We apologize for the quality of Figure 1. We have made efforts to enhance the quality of this Figure, and, in addition, all figures have undergone processing using the software recommended by the editor to ensure their quality.

- Page 4, lines 100-103. The authors should also mention the vaccination as another potential bias to seroprevalence studies.

Thank you for raising this point. In accordance with your suggestion, we have included vaccination as one of the potential sources of bias in seroprevalence studies. 

- Page 9, I assume that L, H and M are mutually exclusive. So the definition of L (working days) should be adapted accordingly. L are the working days excluding Mondays and post-vacation working days

Yes, your comment is right, L, H, and M are mutually exclusive. We appreciate your observation. In the revised version of the article, set L is defined as 'non-Monday, non-post-holiday working days (set L)' both in the main text and in S1 Appendix.

- Fig 6 and 7, could be together.

We have followed your suggestion and they are together in the new version of the paper. 

- Fig 9 and 10 could be together as well.

We attempted to combine both Figures into a single one; however, the resulting figure becomes excessively large if we wish to maintain readability. This can potentially disrupt the layout of the paper. If you still believe that join figure would be more appropriate, we are open to submitting a revised version of the paper accommodating the larger figure.

---

## [Editor Report · Decision Letter 1]

20 Nov 2023

Early detection of new pandemic waves. Control Chart and a new Surveillance Index.

PONE-D-23-10336R1

Dear Dr. Mallor,

We’re pleased to inform you that your manuscript has been judged scientifically suitable for publication and will be formally accepted for publication once it meets all outstanding technical requirements.

Kind regards,

Ralf Reintjes, PhD, MD, MSc(P.H.), MSc(Epi.)

Academic Editor

PLOS ONE
---

## [Editor Report · Acceptance letter]

1 Feb 2024

PONE-D-23-10336R1 

PLOS ONE

Dear Dr. Mallor, 

I'm pleased to inform you that your manuscript has been deemed suitable for publication in PLOS ONE. Congratulations! Your manuscript is now being handed over to our production team.

Kind regards, 

on behalf of

Dr. Ralf Reintjes 

Academic Editor

PLOS ONE